# Rumen-Protected Lysine and Methionine Supplementation Reduced Protein Requirement of Holstein Bulls by Altering Nitrogen Metabolism in Liver

**DOI:** 10.3390/ani13050843

**Published:** 2023-02-25

**Authors:** Songyan Zou, Shoukun Ji, Hongjian Xu, Mingya Wang, Beibei Li, Yizhao Shen, Yan Li, Yanxia Gao, Jianguo Li, Yufeng Cao, Qiufeng Li

**Affiliations:** 1College of Animal Science and Technology, Hebei Agricultural University, Baoding 071000, China; 2College of Animal Medicine, Hebei Agricultural University, Baoding 071000, China

**Keywords:** rumen-protected amino acids, growth performance, nitrogen metabolism, Holstein bulls

## Abstract

**Simple Summary:**

Excessive protein intake causes dietary nitrogen to be excreted through urine nitrogen and fecal nitrogen, reducing nitrogen use efficiency. The main way to reduce dietary nitrogen loss is to reduce dietary protein content, as well as to meet the nutritional needs of ruminants. Therefore, reducing crude proteins while adding rumen amino acids can achieve a reduction in nitrogen emissions. The results showed that adding RPLys (55 g/d) and RPMet (9 g/d) to the bull diet and low protein diet (11%) could improve the growth performance, increase the level of nitrogen metabolism, and enhance the expression of genes related to nitrogen metabolism.

**Abstract:**

The aim of this study was to investigate the effect of low-protein diets supplemented with rumen-protected lysine (RPLys) and methionine (RPMet) on growth performance, rumen fermentation, blood biochemical parameters, nitrogen metabolism, and gene expression related to N metabolism in the liver of Holstein bulls. Thirty-six healthy and disease-free Holstein bulls with a similar body weight (BW) (424 ± 15 kg, 13 months old) were selected. According to their BW, they were randomly divided into three groups with 12 bulls in each group in a completely randomized design. The control group (D1) was fed with a high-protein basal diet (CP13%), while bulls in two low-protein groups were supplied a diet with 11% crude protein and RPLys 34 g/d·head + RPMet 2 g/d·head (low protein with low RPAA, T2) or RPLys 55 g/d·head + RPMet 9 g/d·head (low protein with high RPAA, T3). At the end of the experiment, the feces and urine of dairy bulls were collected for three consecutive days. Blood and rumen fluid were collected before morning feeding, and liver samples were collected after slaughtering. The results showed that the average daily gain (ADG) of bulls in the T3 group was higher than those in D1 (*p* < 0.05). Compared with D1, a significantly higher nitrogen utilization rate (*p* < 0.05) and serum IGF-1 content (*p* < 0.05) were observed in both T2 and T3 groups; however, blood urea nitrogen (BUN) content was significantly lower in the T2 and T3 groups (*p* < 0.05). The content of acetic acid in the rumen of the T3 group was significantly higher than that of the D1 group. No significant differences were observed among the different groups (*p* > 0.05) in relation to the alpha diversity. Compared with D1, the relative abundance of Christensenellaceae_R-7_group in T3 was higher (*p* < 0.05), while that of Prevotellaceae _YAB2003_group and Succinivibrio were lower (*p* < 0.05). Compared with D1 and T2 group, the T3 group showed an expression of messenger ribonucleic acid (mRNA) that is associated with (CPS-1, ASS1, OTC, ARG) and (N-AGS, S6K1, eIF4B, mTORC1) in liver; moreover, the T3 group was significantly enhanced (*p* < 0.05). Overall, our results indicated that low dietary protein (11%) levels added with RPAA (RPLys 55 g/d +RPMet 9 g/d) can benefit the growth performance of Holstein bulls by reducing nitrogen excretion and enhancing nitrogen efficiency in the liver.

## 1. Introduction

Protein, as typically the most expensive macronutrient of diets, plays critical roles in the health, growth, production, and reproduction of animals. However, protein ingredient shortages and nitrogen pollution challenge the livestock farming worldwide, albeit these problems have been alleviated in recent decades due to an increase in demand for animal source food from a fast-growing population with rising incomes [1,2]. Therefore, enhancing the utilization efficiency of dietary protein and reducing excretory losses would be alternative strategies to solve these problems [3].

Low-protein diets have been proven to enhance nitrogen utilization [4,5]. However, restricting N intake also sacrificed the growth performance and productivity of animals [6,7], which has been attributed to limiting amino acid deficiency in low-protein diets [8]. Lysine (Lys) and methionine (Met) are the top two limiting amino acids (LAA) for ruminants [9,10]. Adding rumen-protected Lys and Met in low-protein diets was considered an efficient way to the meet animal amino acids requirement, as they could escape from rumen degradation and increase the supply of amino acids to the intestines, thus improving the N utilization [11]. Incorporating rumen-protected Lys and (or) Met into low-protein diets was reported to increase dry matter intake in transition cows [12,13]. Previous studies also suggested that rumen-protected Lys and (or) Met in low-protein diets promoted milk protein yield in high-producing dairy cows [14,15] and maintained milk production and milk protein yield while reducing the N losses in urine in dairy cows [16]. The question of how to reduce nitrogen emissions of ruminants without affecting their production performance has always been the focus of scholars, and the research in this area has mostly been focused on dairy cows; however, there have been few studies conducted on Holstein bulls.

Nitrogen recycling contributes to effective N utilization in ruminants [17], and ruminal microbiota and the liver play important roles in this nitrogen metabolism [4]. Therefore, the aim of this study was to investigate the effect of low-protein diets supplemented with rumen-protected lysine (RPLys) and methionine (RPMet) on growth performance, rumen fermentation, blood biochemical parameters, nitrogen metabolism, and gene expression related to N metabolism in the livers of Holstein bulls.

## 2. Materials and Methods

This study was conducted between March 2016 and June 2016 at Hongda an animal husbandry in Baoding, P. R. China. The experimental protocol (YXG 1711) was approved by the Institutional Animal Care and Use Committee of Hebei Agricultural University.

### 2.1. Animals, Experimental Design, and Diets

Thirty-six healthy and disease-free Holstein bulls with a similar body weight (BW; 424 ± 15 kg, aged 14 months old) were selected. According to their BW, they were randomly divided into 3 groups with 12 bulls in each group in a completely randomized design. The control group (D1) was fed with a high-protein basal diet (CP13%), while bulls in two low protein groups were supplied diet with 11% crude protein and RPLys 34 g/d·head + RPMet 2 g/d·head (low protein with low RPAA, T2) or RPLys 55 g/d·head + RPMet 9 g/d·head (low protein with high RPAA, T3). Basic diets were prepared according to Japanese feeding standard (2008) for beef cattle [18] (Table 1). The RPAA (Hangzhou Kangdequan Feed Limited Company, Hangzhou, Zhejiang, China) feed was used with a rumen protection rate of 60.0% and was premixed with 100 g of grounded corn which, was used as a carrier for the supplement and was the same amount of grounded corn as that supplied to bulls in the D1 group. All animals were fed ad libitum the basic diets and with free access to clean water. All the experimental animals were housed in tie stalls according to the groups and were fed twice daily at 06:00 and 18:00 h following the removal of the feed refusals before morning feeding. The experiment consisted of 3 periods: a 14-day adaptation period, a 2-month feeding period, and a 7-day sample collection period. Holstein bulls were weighted before morning feeding at the beginning and end of every feeding period.

### 2.2. Sample Collection

The diet offered and refused for individual bulls was weighed every day throughout the trial to average daily dry matter intake (ADMI). Samples of individual feed ingredients, orts, and diets were collected weekly during the experimental period and stored at −20 °C [19]. At the beginning of the experiment, all Holstein bulls were weighed before feeding in the morning to obtain their initial weight. Similarly, at the end of the trial, all Holstein bulls were weighed before morning feeding to obtain the final weight, and the average daily gain (ADG) was calculated as (final weight–initial weight)/test days. Based on the ADMI and ADG, the feed weight ratio (F/G) was calculated. At the end of the feeding period, four Holstein bulls in each group were randomly selected, and a 10-mL blood sample was collected via jugular venipuncture from each bull before morning feeding. The samples were immediately centrifuged at 3000 rpm for 15 min, and the serum samples were collected and stored at −20 °C for further analysis. After 2 h of morning feeding at the end of the feeding period, the ruminal fluid samples of four bulls were collected via an oral stomach tube equipped with a vacuum pump. We discarded the first 100 to 200 mL of fluid collected to reduce the chance that the stomach tube rumen samples were contaminated with saliva. Once again, approximately 200 mL of rumen fluid was collected, and about 20 mL was taken, filtrated with four layers of sterile cheesecloth, and then transferred to 2-mL sterile tubes and stored in liquid nitrogen for further analysis.

Three bulls in each group were randomly selected and euthanized at the end of the feeding experiment after 2 h of morning feeding. The middle part of liver tissue was immediately collected after animal sacrifice and cut into 5-mm fragments; the tissue sample was then placed into sterile tubes and stored in liquid nitrogen for further analysis.

Another three bulls in each group were randomly selected after the feeding period and were transferred to metabolic cages. After a 5-day adaption period, feces and urine were collected during the next 3 days. Total feces and urine were respectively collected daily before morning feeding. The feces of each bull were weighted, mixed, subsampled (100 g/kg), and stored at −20 °C. Each bull fecal sample was evenly divided into two parts, one with 10% (10:1) sulfuric acid solution and the other without acid, before being dried, crushed, sifted, and stored at room temperature for the determination of nutrient content. The urine of each bull was collected using a plastic container with 10 mL of 10% sulfuric acid to prevent the loss of ammonia; then, after the volume was measured, the urine was filtered with four layers of gauze filter, and subsamples (100 mL/individual) were stored at −20 °C for urine nitrogen measurement.

### 2.3. Laboratory Analysis

Offered and refused feed and feces were dried at 55 °C for 48 h, ground to pass through a 1-mm screen (Wiley mill, Arthur H. Thomas, Philadelphia, PA, USA), and stored at 4 °C for analysis of chemical composition. The dry matter (DM, method 934.01), ash (method 938.08), crude protein (CP, method 954.01), ether extract (EE, method 920.39), Ca (method 927.02), and P (method 965.17) contents of the samples were determined according to the procedures of the AOAC [20], and NDF (amylase) and ADF content was analyzed using the methods of Van Soest et al. [21]. Lysine and methionine content in the feed was analyzed using an automatic AA analyzer (Hitachi 835, Tokyo, Japan).

Serum alanine transferase (ALT), aspartate transferase (AST), albumin (ALB), total protein (TP), glucose (GLU), and blood urea nitrogen (BUN) were analyzed using an automatic biochemical analyzer (Hitachi 7020, Tokyo, Japan). Serum growth hormone (GH) and insulin-like growth factor-1 (IGF-1) contents were measured with enzyme-linked immunosorbent assay (ELISA) kits according to the manufacturer’s specifications (HZ Bio.CO., Shanghai, China).

The pH value of the rumen fluid was measured immediately by using a digital pH analyzer (PHS-3C, Shanghai, China), and ammonia nitrogen (NH3-N) and microbial protein (MCP) were determined following recommendations provided in previous studies [22]. Volatile fatty acid (VFA) concentrations in rumen fluid were analyzed using gas chromatography (TP-2060F, Tianpu. Co., Ltd., Beijing, China).

The DNA in rumen fluid was extracted using the CTAB method using a commercial kit (Omega Bio-Tek, Norcross, GA, USA), and, after DNA was purified with 1% agarose gel electrophoresis, the library was constructed using a TruSeq^®^ DNA PCR-Free Sample Preparation Kit (Illumina, Inc., San Diego, CA, USA). Then, the constructed library was quantified using HiSeq2500 PE250 (Illumina, Inc., San Diego, CA, USA). Sequences data were analyzed using the QIIME2 pipeline according to a previous study [23] and submitted to NCBI with project ID P2016030502-S2-3-1.

The primer of target genes (Table 2) was designed according to the bovine gene sequences reported in NCBI and synthesized by the Shanghai Biotedchnology Technology Corporation Limited Company. The total amount of ribonucleic acid (RNA) was extracted from the liver tissue of Holstein bulls with a miRNeasy kit (Qiagen, Hilden, Germany); then, RNA quality was determined using NanoDrop 2000 (NanoDrop Tec, Rockland, DE) with OD260/OD280 ranging between 1.9 and 2.1. Real-time polymerase chain reaction (PCR) was performed to quantify the expression of target genes, using an SYBR Green PCR Master mix (Takara bio-Co., Shiga, Japan) and following the manufacturer’s protocols. The gene expression of liver tissue was calculated using the method of 2-ΔΔCt, where the expression of ACTB was used as referenced D1.

### 2.4. Statistical Analysis

The data management was performed using a spreadsheet program with Excel, and statistical analysis was carried out using R software (version 3.6.3, R Foundation for Statistical Computing, Vienna, Austria.) with a one-way analysis of variance (ANOVA) model: Y = *α* + X*i* + *ei*, where Y is the observed parameters, *α* is the overall mean, X*i* is the *i*th treatment effect, and *ei* is the residual error. All data were shown using least squares means, and significant differences among treatments were declared at *p* < 0.05 and a tendency if 0.05 < *p* ≤ 0.10.

## 3. Results

### 3.1. Growth Performance

There was no significant difference (*p* > 0.05) in ADG, ADMI, and F/G among different groups; however, the F/G in the T2 and T3 groups decreased by 8.45% and 6.67%, respectively, compared with D1 (Table 3).

### 3.2. Nitrogen Metabolism

Compared with the D1 group, the intake of nitrogen and the amount of nitrogen excretion by feces and urine were significantly lower in the T2 and T3 groups (*p* < 0.05). The ratio of nitrogen excretion by feces and nitrogen intake (FN/IN) was lower in T3 compared with the D1 and T2 groups, while the ratio of nitrogen excretion by urine and nitrogen intake (UN/IN) was lower in the T2 and T3 groups compared to the D1 group. Thus, a significantly higher nitrogen utilization rate was observed in both T2 and T3 groups compared with the D1 group (*p* < 0.05; Table 4).

### 3.3. Serum Biochemical Index

Low-protein diet with RPAA supplementation had no effect on concentrations of ALT, AST, ALB, TP, GLU, and GH in serum (*p* > 0.05). Concentration of serum BUN significantly decreased; however, the concentration of serum IGF-1 significantly increased in the T3 group compared with the D1 group (*p* < 0.05; Table 5).

### 3.4. Rumen Fermentation

No significant difference was detected in the rumen pH, concentration of NH3-N, MCP, propionate, and butyrate, and in the ratio of acetate/propionate among different groups (*p* > 0.05). The concentration of acetate in the T3 group was significantly higher than that in D1 and T2 (*p* < 0.05; Table 6).

### 3.5. Rumen Microbiota

No significant difference was observed in alpha diversity among the different groups (*p* > 0.05; Table 7). The relative abundance of the highest 16 abundant bacteria at the genus level was compared among the different groups. However, the relative abundance of Ruminococcaceae_NK4A214 in the T3 group was lower than that in the D1 group (*p* < 0.05), and the abundance of Christensenellaceae_R-7_group in the T3 group was lower than that in both D1 and T2 groups (*p* < 0.05). Meanwhile, the relative abundance of Prevotellaceae_YAB2003_group in T3 was higher than that in the D1 group (*p* < 0.05), and the relative abundance of Succinivibrio in T3 was higher than that in both the D1 and T2 groups (*p* < 0.05; Table 8).

### 3.6. Gene Expression in Liver Tissue

The expression of the CPS-1, ASS, ARG, OTC, and N-AGS genes, which relate to nitrogen metabolism or urea metabolism in liver tissue, are shown in Figure 1. The expression of CPS-1, ARG, and N-AGS was significantly upregulated in the T3 group (*p* < 0.05), although no significant difference was observed between the rT2 and D1 groups (*p* > 0.05). The expression of CPS-1, ARG, and N-AGS increased by 25%, 18%, and 13% in the T2 group compared with D1. The expression of ASS and OTC was upregulated in both the T2 and T3 groups compared with D1 (*p* < 0.05).

The expression of the SLC3A2, IRS1, PDK, P13K, TSC1, TSC2, mTORC1, eIF4EBP1, S6K1, and eIF4B genes, which are related to the nitrogen metabolism in liver tissue, are shown in Figure 2. The low-protein diet with RPAA supplementation did not affect gene expression of SLC3A2, P13K, TSC2, and eIF4EBP1 (*p* > 0.05); however, the expression of IRS1, PDK, S6K1, and eIF4B genes in liver tissue increased significantly (*p* < 0.05), and the expression of the mTORC1 gene also increased (*p* = 0.09), while the expression of TSC1 gene decreased significantly (*p* < 0.05).

## 4. Discussion

Protein is one major factor that affects the health, growth, and production of ruminants. Moreover, although people tend to formulate high-protein diets to achieve a better production of ruminants, the global protein shortage is increasing [1], and high-protein diets overload the environment by increasing nitrogen (N) excretion through urine and feces [3], which is harmful for the sustainability of the livestock industry.

By providing bulls with a low-protein diet (11% CP) supplemented with rumen-protected lysine and methionine, our findings indicate that, compared with a high-protein diet (13% CP) group which followed the recommended Japanese feeding standard for beef cattle [18], our low-protein diet supplemented with RPAA increased ADG and N utilization and decreased N excretion through urine and feces. These findings were comparable with previous studies in which the feeding of rumen-protected Lys and (or) Met to castrated cattle increased daily gain [24] and reduced urinary nitrogen and urea nitrogen in urine [25]. The World Health Organization (WHO) proved that the addition of RPAA to a low-protein diet increases N utilization, reduces N emission and environmental pollution, and promotes the growth performance of dairy cows [12,14].

Blood biochemical parameters are sensitive to animal health and nutrient condition [26,27]. The serum content of ALT, AST, ALB, TP, GLU, BUN, GH, and IGF-1 was used to assess the nutrient condition of bulls with different treatment groups. From this, we observed that BUN content decreased, and IGF-1 content increased, in bulls provided with a low-protein diet supplemented with RPAA, while other indexes were not affected. The serum BUN content reflects the nitrogen balance of ruminants and negatively correlated with N utilization [17]. When ruminants were provided with low-dietary protein with a higher N utilization, serum BUN decreased [4,28]. The main function of IGF-1 relates to the inhibiting of protein degradation and the promoting of protein synthesis to maintain nitrogen balance and to improve the growth performance of animals [29,30]. These observations further explained the improvement in N utilization and growth performance of bulls on a low-protein diet supplemented with RPAA.

When cattle are fed with low-protein diets, urea N recycling can be considered a high-priority metabolic function because a continuous N supply for microbial growth in the rumen is a strategy for animal survival [31]. The abundance of the microflora reflects its ability to adapt to a particular environment and compete for available nutrients; moreover, it indicates its importance to the overall function of the microbiome as a whole [32]. The ACE (reflecting the richness of bacteria in the sample), Shannon, and PD-whole-tree (reflecting the microbial diversity in feces) indexes were used to assess the alpha diversity of rumen microbiota. Previous studies have demonstrated that rumen fermentation and microbiota are sensitive to protein levels [33,34] or feed ingredients [35] in ruminants, which were also sensitive biomarkers of N utilization [36]. By monitoring the rumen fermentation and microbiota, we observed an increase in the acetate content of rumen; however, other parameters including NH3-N and MCP content were not significant affected, which is similar to the results of a study by Martin et al. [37]. The addition of methionine analogue 2-hydroxy-4-methylthiobutyric acid (HMB) and esterified 2-hydroxy-4-methylthiobutyric acid (HMBi) to the diet of dairy cows significantly increased the content of rumen total volatile fatty acids (TVFAs) (37). Some studies have shown that methionine hydroxy analogue (MHA) can increase the ratio of acetic acid and butyric acid in rumen content [38]. Research has showed that 0.52% of methionine could increase the content of butyric acid in rumen, while 0.26% methionine did not affect the content of VFA [39]. The above results show that the effect of methionine on rumen VFA content is unpredictable. The alpha diversity of microbiota in rumen was not affected by treatment, and only a small portion of bacteria at the genus level (~5% in abundance) was determined to be significantly different between groups with a decreased relative abundance of Ruminococcaceae_NK4A214_group and Christensenellaceae_R-7_group and increased Prevotellaceae_YAB2003_group and Succinivibrio in bulls on a low-protein diet supplemented with RPAA. These findings hinted that bulls on a low-protein diet supplemented with RPAA would maintain the rumen fermentation and maintain ruminal microbiota homeostasis compared with that from D1.

The liver plays important roles in the utilization efficiency of recycled N. The excess nitrogen in the rumen is usually inhaled into the animal’s blood in the form of ammonia, which is then metabolized by the liver to synthesize urea. All the urea synthesized by the liver, some of which is secreted via saliva into the rumen and intestines of animals, are reused by bacteria, protozoa, and other microorganisms; the other part is filtered by the kidneys and excreted with the urine [28]. The urea cycle plays a key role in maintaining a positive balance of nitrogen in anima, especially at low dietary nitrogen levels. S6K1 and eIF4EBP1 are genes that regulate protein translation downstream of mTORC1. The S6K1 gene can promote protein translation by stimulating the phosphorylation of downstream eIF-4B, RPS6, eIF-2, and PAPB [40], and the SLC3A2, IRS1, PDK, P13K, TSC1, TSC2, mTORC1, eIF4EBP1, S6K1, and eIF4B genes are related to nitrogen metabolism in the liver; moreover, these genes would become overexpressed when blood ammonia increased to increase urea synthesis and balance the blood ammonia [41]. However, unexpected results were observed in the current experiment: when feeding bulls with a low-protein diet supplemented with RPAA, we observed that the serum BUN decreased but the expression of genes associated with urea synthesis in liver increased. This finding can explain why the low-protein diet supplemented with RPAA induced an increase in N efficiency; however, the mechanism behind these upregulated genes in the liver was unclear. Previous studies have demonstrated that AA in diets not only provide animal nutrition but also act as a functional regulator and have ability to stimulate expression altering in multiple tissue cells such as mammary tissue [42], polymorphonuclear cells [43], and adipose tissue [44], as well as liver tissue [45,46]. The influence of RPLys and RPMet on liver genes’ expression requires further study. As the number of samples selected in this study is limited, it is necessary to further test the current data in the future research.

## 5. Conclusions

In summary, providing low dietary protein (11%) with RPLys (55 g/d) and RPMet (9 g/d) to bulls could increase their nitrogen utilization rate, serum IGF-1 content, ruminal acetate content, and expression genes associated with urine metabolism and nitrogen metabolism in liver compared to that with high protein (13%). Our findings indicate that providing a low-protein diet supplemented with RPAA could benefit bulls mainly by increasing liver nitrogen metabolism and utilization; however, the RPAA’s affecting of liver gene expression at a nutrition level or as a signal molecule still requires further study.

## Figures and Tables

**Figure 1 animals-13-00843-f001:**
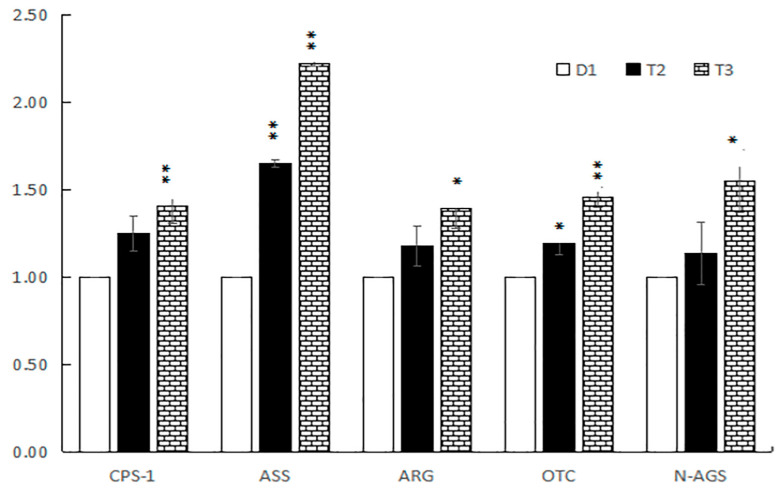
The relative expression levels of CPS-1, ASS, ARG, OTC, and N-AGS mRNA in liver. Control group (D1), low protein with low RPAA (T2), and low protein with high RPAA (T3). *CPS-1* = Carbamoyl-phosphate synthase 1; *ASS* = Argininosuccinate synthase; *ARG* = Arginase; *OTC* = Ornithine carbamoyltransferase; *N-AGS* = N-acetylglutamate synthase. *: The difference is significant. **: The difference is extremely significant.

**Figure 2 animals-13-00843-f002:**
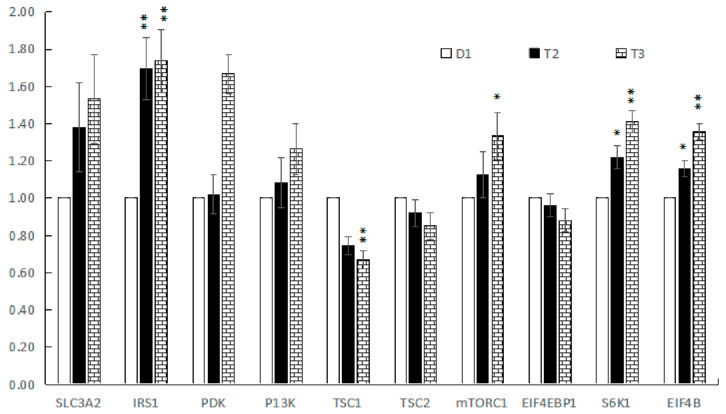
The relative expression levels of SLC3A2, IRS1, PDK, P13K, mTORC1, eIF4EBP1, S6K1, and eIF4B mRNA in liver. Control group (D1), low protein with low RPAA (T2), low protein with high RPAA (T3). SLC3A2 = Solute carrier family 3 (amino acid transporter heavy chain), member 2; IRS1 = Insulin receptor substrate 1; PDK = Phosphoinositide-dependent protein kinase; P13K = Phosphoinositol 3-kinase; TSC1 = Tuberous sclerosis complex 1; TSC2 = Tuberous sclerosis complex 2; mTORC1 = Mammalian target of rapamycin complex 1; S6K1 = Ribosomal protein S6 kinases; eIF-4B = Eukaryotic initiation factor. *: The difference is significant. **: The difference is extremely significant.

**Table 1 animals-13-00843-t001:** Ingredients and chemical compositions of experimental diets.

Items ^2^	Treatment ^1^		
	D1	T2	T3
Ingredient, % of DM			
Flaked corn (%)	35.72	37.7	37.7
Bran (%)	0	3.72	3.72
Soybean meal (%)	4.2	1	1
Cottonseed meal (%)	7.5	3.8	3.8
DDGS (%)	4	5.2	5.2
NaHCO_3_ (%)	0.83	0.83	0.83
Premix ^2^ (%)	2.75	2.75	2.75
RPLys (g/d)		34	55
RPMet (g/d)		2	9
Corn straw silage (%)	36	36	36
Millet straw (%)	9	9	9
Total (%)	100	100	100
Chemical composition			
CP (%)	13.00	11.00	11.00
Ca (%)	0.73	0.73	0.73
P (%)	0.36	0.35	0.35
NDF (%)	38.59	39.25	39.25
ADF (%)	23.11	22.82	22.82
Lys (%)	0.24	0.40	0.51
Met (%)	0.14	0.14	0.18
NEmf ^3^ (MJ/kg)	6.42	6.42	6.42
MP (%)	10.03	9.09	9.09

^1^ (1) control group (D1), (2) low protein with low RPAA (T2), (3) low protein with high RPAA (T3). ^2^ The premix provided the following per kg of feed: VA 8 800 IU, VD_3_ 2 035 IU, VE 82.5 mg, D-biotin 1.1 mg, Nicotinamide 55 mg, β–carotene 3.3 mg, Cu 18.7 mg, Mn 49.5 mg, Zn 82.5 mg, Mg 880 mg, Co 0.66 mg, I 1.1 mg, Se 0.55 mg, Ethoxyquin 13.75 mg. ^3^ NE_mf_ is a calculated value, while the other nutrient levels are measured values.

**Table 2 animals-13-00843-t002:** Primer of target genes in liver.

Gene Name	Accession Number	Primer and Probe Sequences	Amplification Products
Mammalian target of rapamycin complex 1 (*mTORC1*)	XM_002694043.5	Forward: AAGCCGCGCGAACCT	177
		Reverse: CTCCATGGTGACGTAGTGCT	
Insulin receptor substrate 1 (*IRS1*)	XM_003585773.4	Forward: CTCCAGCGAGGATCTAAGCG	100
		Reverse: TAGGTCTTCATTCTGCTGTGATGT	
Phosphoinositol 3-kinase (*P13K*)	NM_001206047.1	Forward: CTTATTTCAAGAGCCCCGTGGT	115
		Reverse: TGGAGCCATCAGACGCTATC	
Phosphoinositide-dependent protein kinase (*PDK*)	NM_001205957.1	Forward: TGAGAGCAACGATGGAGCAT	107
		Reverse: TCCTCGGTCACTCATCTTCAC	
Tuberous sclerosis complex 1 (*TSC1*)	XM_005213449.3	Forward: CCTGTCTACACAGACAACCCC	157
		Reverse: ATGGCCGCTTCTTCTTTATCCT	
Tuberous sclerosis complex 2 (*TSC2*)	XM_010819198.2	Forward: AGAAGAGCTGGCGGACTTT	120
		Reverse: GATGCGATGTACTCATCAAGGT	
Eukaryotic initiation factor (*eIF-4B*)	XM_005206207.1	Forward: CTGGAAGGGGATGTTTCAACC	116
		Reverse: ATATTGGGTTCCCGAGCAGC	
Eukaryotic initiation factor 4E binding protein 1 (*eIF-4EBP1*)	NM_001077893.2	Forward: CGGGGTCACTAGCCCTACA	105
		Reverse: AACTGTGACTCTTCACCGCCTG	
Ribosomal protein S6 kinases (*S6K1*)	NM_205816.1	Forward: CAACCAGGTCTTTCTGGGTT	121
		Reverse: TGTTCGTGGGCTGCCAATAA	
Solute carrier family 3 (amino acid transporter heavy chain), member 2 (*SLC3A2*)	NM_001024488.2	Forward: GCAACCTAGCGGACCTAAAGGA	109
		Reverse: GTCTCTGTGAGGTCATCCTCC	
Carbamoyl-phosphate synthase 1 (*CPS-1*)	XM_015458817.1	Forward: TGGGATTAAGGTTGCAGGTTTG	100
		Reverse: CTTTTCTTCCCGTAGCCACT	
N-acetylglutamate synthase (*N-AGS*)	XM_002696039.4	Forward: CTCTTCAGCAACAGGGGTTC	141
		Reverse: GTAGTCGTCCCGGAGCTTTT	
Ornithine carbamoyltransferase (*OTC*)	NM_177487.2	Forward: AGTGCGGCTAAATTCGGGAT	113
		Reverse: AGCTTGGTACCGTTCTCCTTG	
Argininosuccinate synthase (*ASS*)	NM_173892.4	Forward: CCACAGGAAAGGGGAACGAC	100
		Reverse: TAGAACTCGGGCATCCTCCA	
Arginase (*ARG*)	XM_005210924.3	Forward: AGAACTAGAGTGTGATGTGAAAGA	122
		Reverse: CAGCCAGCTTTTCACTTGCT	

**Table 3 animals-13-00843-t003:** Effects of low-protein dietary inclusion of RPLys and RPMet on average daily gain, average daily feed intake, and feed/gain ratio of Holstein bulls.

Items ^2^	Treatment ^1^			*p*-Value
	D1	T2	T3	
ADG (kg)	1.37 ± 0.22	1.42 ± 0.06	1.55 ± 0.07	0.055
ADMI (kg)	10.09 ± 1.20	10.17 ± 0.83	10.09 ± 0.81	0.967
F/G	7.34 ± 1.05	7.20 ± 0.80	6.72 ± 0.75	0.157

^1^ (1) Control group (D1), (2) low protein with low RPAA (T2), (3) low protein with high RPAA (T3). ^2^ ADG = average daily gain; DMI = dry matter intake; ADMI = daily dry matter intake; F/G = feed weight ratio.

**Table 4 animals-13-00843-t004:** Effects of low-protein dietary inclusion of RPLys and RPMet on nitrogen metabolism of Holstein bulls.

Items ^2^	Treatment ^1^			*P*-Value
	D1	T2	T3	
Intake of Nitrogen (IN) (g/d/·head)	216.76 ± 3.49 ^a^	181.81 ± 0.40 ^b^	179.69 ± 1.16 ^b^	<0.001
Fecal Nitrogen(FN) (g/d·head)	71.10 ± 0.42 ^a^	59.98 ± 0.76 ^b^	57.50 ± 0.23 ^c^	<0.001
Urinary Nitrogen (g/d·head)	62.76 ± 1.20 ^a^	45.74 ± 2.89 ^b^	43.31 ± 0.38 ^b^	<0.001
FN/IN (%)	32.81 ± 0.48 ^a^	32.99 ± 0.39 ^a^	32.00 ± 0.17 ^b^	0.037
UN/IN (%)	28.95 ± 0.33 ^a^	25.16 ± 1.59 ^b^	24.10 ± 0.17 ^b^	0.002
Nitrogen Utilization Rate (%)	38.24 ± 0.70 ^b^	41.85 ± 1.97 ^a^	43.90 ± 0.31 ^a^	0.004
Nitrogen Metabolic Rate (%)	56.91 ± 0.69 ^b^	62.45 ± 0.31 ^a^	64.55 ± 0.31 ^a^	0.002

^1^ (1) Control group (D1), (2) low protein with low RPAA (T2), (3) low protein with high RPAA (T3). ^2^ Note: nitrogen utilization rate (%) = 100 × (IN − FN − UN)/IN; nitrogen metabolic rate (%) = 100 × digestion nitrogen rate/nitrogen utilization rate. a, b, c Means within a row with different superscripts differ (*p* ≤ 0.05).

**Table 5 animals-13-00843-t005:** Effects of low-protein dietary inclusion of RPLys and RPMet on blood biochemistry of Holstein bulls.

Items ^2^	Treatment ^1^			*p*-Value
	D1	T2	T3	
ALT(U/L)	23.95 ± 1.30	24.68 ± 0.39	25.49 ± 6.03	0.833
AST(U/L)	70.78 ± 3.80	71.01 ± 2.34	73.00 ± 6.50	0.758
ALB(g/L)	23.79 ± 0.68	23.89 ± 1.50	24.32 ± 1.52	0.872
TP(g/L)	62.94 ± 1.27	62.97 ± 1.23	64.47 ± 1.24	0.196
GLU(mmol/L)	3.35 ± 0.14	3.44 ± 0.14	3.49 ± 0.25	0.542
BUN(mmol/L)	2.80 ± 0.50 ^a^	2.31 ± 0.24 ^ab^	1.81 ± 0.09 ^b^	0.026
GH(μg/L)	22.66 ± 4.53	22.99 ± 1.84	23.16 ± 2.14	0.980
IGF-1(μg/L)	142.91 ± 1.74 ^b^	144.47 ± 3.36 ^ab^	149.32 ± 2.14 ^a^	0.047

^1^ (1) Control group (D1), (2) low protein with low RPAA (T2), (3) low protein with high RPAA (T3). ^2^ ALT = serum alanine transferase; AST = aspartate transferase; ALB = albumin; TP = total protein; GLU = glucose; BUN = blood urea nitrogen; GH = serum growth hormone; IGF-1 = insulin-like growth factor-1. a, b Means within a row with different superscripts differ (*p* ≤ 0.05).

**Table 6 animals-13-00843-t006:** Effects of low-protein dietary inclusion of RPLys and RPMet on rumen fermentation of Holstein bulls.

Items ^2^	Treatment ^1^			*p*-Value
	D1	T2	T3	
pH	6.50 ± 0.21	6.39 ± 0.05	6.44 ± 0.13	0.598
NH_3_-N(mg/dL)	12.67 ± 1.34	12.29 ± 3.03	12.23 ± 3.35	0.978
MCP(mg/mL)	4.59 ± 1.37	4.91 ± 0.41	4.95 ± 0.77	0.843
Acetate(mmol/L)	86.35 ± 6.65 ^b^	87.56 ± 0.94 ^b^	95.54 ± 1.30 ^a^	0.026
Propionate(mmol/L)	36.92 ± 2.52	37.84 ± 2.91	38.01 ± 0.36	0.818
Butyrate(mmol/L)	14.13 ± 1.85	14.28 ± 0.43	13.32 ± 0.01	0.546
Acetate/Propionate (A/P)	2.41 ± 0.11	2.32 ± 0.20	2.40 ± 0.18	0.801

^1^ (1) Control group (D1), (2) low protein with low RPAA (T2), (3) low protein with high RPAA (T3). ^2^ NH_3_-N = ammonia nitrogen; MCP = microbial protein; VFA = volatile fatty acid (VFA). a, b Means within a row with different superscripts differ (*p* ≤ 0.05).

**Table 7 animals-13-00843-t007:** The effects of RPLys and RPMet on the alpha diversity analysis index of Holstein bulls.

Items		Treatment ^1^			*p*-Value
		D1	T2	T3	
Richness index	Shannon	8.37 ± 0.57	8.53 ± 0.37	8.09 ± 0.49	0.462
Diversity index	ACE	2178.57 ± 281.67	2063.77 ± 81.19	1942.29 ± 90.40	0.224
Phylogenetic diversity	PD-whole-tree	94.83 ± 5.33	95.79 ± 5.48	88.50 ± 6.73	0.218

^1^ (1) Control group (D1), (2) low protein with low RPAA (T2), (3) low protein with high RPAA (T3).

**Table 8 animals-13-00843-t008:** Effects of low-protein dietary inclusion of RPLys and RPMet on bacterial genus in the rumen fluids of Holstein bulls.

Items	Treatment ^1^			*p*-Value
	D1	T2	T3	
Prevotella_1	23.78 ± 0.11	23.16 ± 7.20	27.72 ± 6.56	0.589
Rikenellaceae_RC9_gut_group	5.39 ± 0.22	4.87 ± 0.89	3.87 ± 1.33	0.209
Succinivibrionaceae_UCG-002	4.59 ± 2.09	3.92 ± 0.98	3.87 ± 1.08	0.805
unidentified_Bacteroidales_RF16_group	3.88 ± 1.19	2.67 ± 0.65	3.28 ± 1.49	0.380
Ruminococcaceae_NK4A214_group	2.61 ± 0.42 ^a^	2.22 ± 0.63 ^ab^	1.40 ± 0.52 ^b^	0.030
Ruminococcaceae_UCG-014	2.48 ± 0.51	2.50 ± 0.38	2.44 ± 0.78	0.990
Ruminobacter	2.28 ± 0.66	3.35 ± 2.64	2.04 ± 0.47	0.495
Christensenellaceae_R-7_group	2.04 ± 0.36 ^a^	1.85 ± 0.53 ^a^	1.02 ± 0.41 ^b^	0.022
UGG003 Prevotellaceae_UCG-003	1.80 ± 0.44	2.52 ± 0.98	1.78 ± 0.53	0.275
Treponema_2	1.52 ± 0.13	1.64 ± 0.31	1.93 ± 0.38	0.290
Fibrobacter	1.42 ± 0.44	1.68 ± 0.35	1.23 ± 0.24	0.360
Ruminococcus_1	0.76 ± 0.13	0.70 ± 0.20	0.57 ± 0.1	0.301
Selenomonas_1	0.55 ± 0.11	0.53 ± 0.27	0.57 ± 0.07	0.958
Succinimonas	0.55 ± 0.15	0.48 ± 0.16	0.35 ± 0.03	0.236
Prevotellaceae_YAB2003_group	0.27 ± 0.02 ^b^	0.38 ± 0.14 ^ab^	0.59 ± 0.05 ^a^	0.011
Succinivibrio	0.20 ± 0.04 ^b^	0.26 ± 0.03 ^b^	0.53 ± 0.08 ^a^	0.001

^1^ (1) Control group (D1), (2) low protein with low RPAA (T2), (3) low protein with high RPAA (T3). a, b Means within a row with different superscripts differ (*p* ≤ 0.05).

## Data Availability

The data presented in this study are available on request from the corresponding author.

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
