# Peer review of "Rumen-Protected Lysine and Methionine Supplementation Reduced Protein Requirement of Holstein Bulls by Altering Nitrogen Metabolism in Liver"

_animals, 2023, doi:10.3390/ani13050843_

Round 1

Reviewer 1 Report

This paper contains valuable information for beef cattle farming. Also, the manuscript is well written and the quality of the paper is high. Please check the following points.

L56, 58 dietary => diet

L130, 147 urea => urine

L145 Use => use

L152 serum insulin-like growth hormone => serum growth hormone

L153 and growth factor-1 => and insulin-like growth factor-1

L164 Did you use HiSeq instead of MiSeq for microbiota analysis?

L241 urine metabolism => nitrogen metabolism or urea metabolism

L280-281 This sentence makes no sense.. Please reconsider.

This study revealed that feeding lysine and methionine, the amino acids necessary for cattle growth, could reduce protein in the diet. This study is also excellent in that it examines nitrogen excretion and hepatic nitrogen metabolism in addition to growth performance. This research method has been carried out with purpose, and the discussion and conclusions of this manuscript are based on the experimental results.  

For references, the following previous research on Holstein steers may be helpful. 1) Effects of low‐crude protein diets supplemented with rumen‐protected lysine and methionine on fattening performance and nitrogen excretion of Holstein steers https://doi.org/10.1111/asj.13562 2) Feedlot performance and carcass characteristics of Holstein steers as affected by source of dietary protein and level of ruminally protected lysine and methionine https://doi.org/10.2527/1995.73123503x 3) The Limiting Amino Acids in Growing Cattle https://doi.org/10.2527/jas1978.463740x

Author Response

Dear Reviewer: We are truly grateful to your critical comments and thoughtful suggestions again for our manuscript. They are really helpful and based on these comments and suggestions, we have revised the manuscript carefully. Revised portions have been marked in red in the revised manuscript. In the following pages are our point-by-point responses to your comments/suggestions.

Comment 1:L56, 58 dietary => diet

Response 1:

Thank you for your suggestion. According to your suggestion, we have revised in the revised manuscript. The sentence “ dietary” has been revised to “diet”.(Line 56, 58)

Comment 2:L130, 147 urea => urine 

Response 1:

Thank you for your suggestion. According to your suggestion, we have revised in the revised manuscript. The sentence “ urea” has been revised to “urine”.(Line 56, 58)

Comment 3:L145 Use => use

Response 1:

Thank you for your suggestion. According to your suggestion, we have revised in the revised manuscript. The sentence “ Use” has been revised to “use”.(Line 143)

Comment 4:L152 serum insulin-like growth hormone => serum growth hormone

Response 1:

Thank you for your suggestion. According to your suggestion, we have revised in the revised manuscript. The sentence “ serum insulin-like growth hormone” has been revised to “serum growth hormone”(Line 148)

Comment 5:L153 and growth factor-1 => and insulin-like growth factor-1

Response 5:

Thank you for your suggestion. According to your suggestion, we have revised in the revised manuscript. The sentence “ and growth factor-1” has been revised to “and insulin-like growth factor-1”(Line 149)

Comment 6:L164 Did you use HiSeq instead of MiSeq for microbiota analysis?

Response 6:

Yes, we used HiSeq for microflora analysis.

Comment 7:L241 urine metabolism => nitrogen metabolism or urea metabolism

Response 7:

Thank you for your suggestion. According to your suggestion, we have revised in the revised manuscript. The sentence “ urine metabolism” has been revised to “itrogen metabolism or urea metabolism”(Line 238,239)

Comment 8:L280-281 This sentence makes no sense. Please reconsider.

Response 8:

Thank you for asking this question. who demonstrated that providing low protein diet with supplementation of RPAA benefited dairy cows mainly on the N utilization increasement, The sentence has been revised to” World Health Organization (WHO) proved that the addition of RPAA to low-protein diet mainly increased N utilization, reduced N emission, reduced environmental pollution, and promoted the growth performance of dairy cows.” in the revised manuscript. (Line 282-284)

Reviewer 2 Report

This paper adds to the knowledge bank regarding rumen protected amino acids supplemented to Holstein Bulls. Overall the paper has some quality information for producers as well as more intense researchers. 

Once reading the paper, there are a few questions I have, especially regarding the supplement. What is the basis of using 11% as the "low protein diet"? In the Materials & Methods, how old are the Holstein bulls? Why were bulls only weighed at the beginning and end of the supplementation stage?

Given as you had only 12 bulls / treatment, why were only 4 bulls of each group chosen for blood season? Also, were the 3 bulls from each treatment group that rumen fluid was collected from, part of the subset blood was collected from? Were any other measurements / samples taken from the 3 bulls of each group which were harvested or just the liver samples? Was there a reason why both rumen protected amino acids were used, as opposed to evaluation of supplementing each separately?

- For the Laboratory Analyses (2.3; lines 140 - 175), are there citations for these procedures?

- In the results section, be sure all of your "p - values" are consistent in being represented as either P or p.

- For tables 3 - 8, be sure that the items in the legend at the bottom are correlated with a superscript in the table. For example, treatments are listed in the legend, but nothing is tagged with a superscript in the table.

- For figures 1 & 2, denote what the 1 or 2 asterisks mean for the research.

- Line 241, is "urine" the correct word?

- Line 280, there seems to be part of this sentence which is missing, please double check.

- Line 306, please double check reference as it is listed different between the paper and the Citations.

- Double check citations for mistakes of spelling, spacing, grammar.

Author Response

Dear Reviewer: We are truly grateful to your critical comments and thoughtful suggestions again for our manuscript. They are really helpful and based on these comments and suggestions, we have revised the manuscript carefully. Revised portions have been marked in red in the revised manuscript. In the following pages are our point-by-point responses to your comments/suggestions.

Comment 1: Once reading the paper, there are a few questions I have, especially regarding the supplement. What is the basis of using 11% as the "low protein diet"? 

Response 1:

Low-protein diet refers to reducing the level of crude protein in the diet by 2% Mur4% in accordance with the standards recommended by NRC, and then adding an appropriate amount of synthetic amino acids to meet the needs of animals for amino acids, so that the type, proportion and quantity of amino acids in the diet can meet the needs of livestock and poultry [1]. At present, there are many and in-depth studies on low-protein diets under the amino acid balance model of pigs and chickens, but relatively few studies on ruminants, so referring to the results of monogastric animals. Therefore, in this experiment, the dietary protein level of the control group was 13%, and the diet of the experimental group was reduced by 2% on the basis of the control group.

Reference:

[1] Le Bellego L, van Milgen J, Dubois S, et al. Energy utilization of low-protein diets in growing pigs[J]. Journal of Animal Science, 2001, 79(5): 1259-1267.

Comment 2: In the Materials & Methods, how old are the Holstein bulls?

 Response 2:

Thirty-six Holstein bulls with similar body weight (BW,424 ± 15 kg, ages,14 months old), healthy and disease-free were selected.(Line 81)

Comment 3:Why were bulls only weighed at the beginning and end of the supplementation stage?

Response 3:

The experimental period is about 3 months. The main purpose of this experiment is to compare the effect of daily weight gain of Holstein bulls between the unsupplemented amino acid group and the supplemental amino acid group. During the experimental period, in order to avoid external stress reaction, only the initial and late stages of weighing were carried out.

Comment 4:Given as you had only 12 bulls / treatment, why were only 4 bulls of each group chosen for blood season? Also, were the 3 bulls from each treatment group that rumen fluid was collected from, part of the subset blood was collected from? Were any other measurements / samples taken from the 3 bulls of each group which were harvested or just the liver samples?

Response 4:

When sampling cattle, healthy experimental cattle with similar body weight, physical condition and good mental state were selected. When sampling cattle were selected, healthy cattle with similar body weight, body condition and good mental state were selected in each treatment group, and all blood samples were collected from 4 cattle selected in each group. Rumen fluid and liver samples were collected from 3 cattle in each treatment group.

Comment 5:Was there a reason why both rumen protected amino acids were used, as opposed to evaluation of supplementing each separately?

Response 5:

At present, researchers at home and abroad have conducted in-depth studies on the addition of rumen lysine and rumen methionine alone, and the results of positive production response, no response or even negative response have been reported [1,2]. Rumen protective methionine and lysine supplementation can increase milk protein yield [3], and the effects of rumen protective amino acids on the production performance of dairy cows are not consistent [4]. The cause of the difference has not been fully studied, which may be related to the level of addition and different experimental animals [5-6]. therefore, it is feasible to design the combination of low-dose and high-dose amino acids in this experiment.

[1] Bauman, D. E. 1999. Bovine somatotropin and lactation: from basic science to commercial application. Domestic Animal Endocrinology 17(2–3):101-116.

[2] Socha, M. T., D. E. Putnam, B. D. Garthwaite, N. L. Whitehouse, N. A. Kierstead, C. G.

Schwab, G. A. Ducharme, and J. C. Robert. 2005. Improving intestinal amino acid supply of pre- and postpartum dairy cows with rumen-protected methionine and lysine. J Dairy Sci. 88(3):1113-1126.

[3] Patton, R. A. 2010. Effect of rumen-protected methionine on feed intake, milk production,true milk protein concentration, and true milk protein yield, and the factors that influence these effects: a meta-analysis. Journal of dairy science 93(5):2105-2118.

[4] Robinson, P. H. 2010. Impacts of manipulating ration metabolizable lysine and methionine levels on the performance of lactating dairy cows: a systematic review of the literature. Livestock Science 127(2–3):115-126.

[5] Kazumasa WATANABE, Alan H FREDEEN, Peter H ROBINSON, et al. Effects of fat coated rumen bypass lysine and methionine on performance of dairy cows fed a diet deficient in lysine and methionine[J]. Animal Science Journal, 2006, 77(5): 495-502.

[6] C. Lee, F Giallongo, A N Hristov, et al. Effect of dietary protein level and rumen-protected amino acid supplementation on amino acid utilization for milk protein in lactating dairy cows[J]. Journal of Dairy Science, 2015, 98(3): 1885-1902.

Comment 6:For the Laboratory Analyses (2.3; lines 140 - 175), are there citations for these procedures?

Response 6:

Thank you for your question, the nutritional composition determination part of the self-determination, blood, genes and other indicators submitted by the company for testing. According to your suggestion, the cited literature has been added to the revised draft.

Reference:

National Academy Press: Washington, DC, USA, 2016. AOAC. Official Methods of Analysis, 17th ed.; Association of Official Analytical Chemist: Arlington, VA, USA, 2008.

Van Soest, P. J.; Robertson, J. B.; Lewis, B.A. Methods for dietary fiber, neutral detergent fiber, and nonstarch polysaccharides in relation to animal nutrition. J. Dairy Sci. 1991, 74, 3583-3597. https://doi.org/10.3168/jds.S0022-0302(91)78551-2.

Comment 7:In the results section, be sure all of your "p - values" are consistent in being represented as either P or p.

Response 7:

Thank you for your suggestion. According to your suggestion, The sentence “P - values” has been revised to “p - value” in the revised manuscript.

Comment 8:For tables 3 - 8, be sure that the items in the legend at the bottom are correlated with a superscript in the table. For example, treatments are listed in the legend, but nothing is tagged with a superscript in the table.

Response 8:

Thank you for your advice. According to your suggestion, we have added the legend associated with the table to the table and to the modified manuscript.(tables 3 - 8)

Comment 9: For figures 1 & 2, denote what the 1 or 2 asterisks mean for the research.

Response 9:

Thank you for your advice. According to your suggestion, we have added the meaning of 1 or 2 asterisk to the revised manuscript.( figures 1 & 2)

Comment 10:Line 241, is "urine" the correct word?

Response 10:

Thank you for your suggestion. According to your suggestion, The sentence “urine” has been revised to “urea” in the revised manuscript. (Line 240)

Comment 11: Line 280, there seems to be part of this sentence which is missing, please double check.

Response 11:

Thank you for asking this question. who demonstrated that providing low protein diet with supplementation of RPAA benefited dairy cows mainly on the N utilization increasement, The sentence has been revised to” World Health Organization (WHO) proved that the addition of RPAA to low-protein diet mainly increased N utilization, reduced N emission, reduced environmental pollution, and promoted the growth performance of dairy cows.” in the revised manuscript. (Line 282-284)

Comment 12:Line 306, please double check reference as it is listed different between the paper and the Citations.

Response 12:

Thank you for asking that question. I apologize for the inconvenience, because I didn't express it clearly in my writing. The sentence “By monitoring the rumen fermentation and microbiota, we observed acetate content increased in rumen, but other parameters including NH3-N and MCP content were not significant affected, which is similar to the results of a study by Martina et al.” has been revised to “By monitoring the rumen fermentation and microbiota, we observed acetate content increased in rumen, but other parameters including NH3-N and MCP content were not significant affected, which is similar to the results of a study by Martin et al.” in the revised manuscript.(Line 306)

Comment 13:Double check citations for mistakes of spelling, spacing, grammar.

Response 13:

Thank you for your suggestion. According to your suggestion, we checked the spelling and other problems in the quotation.

Round 2

Reviewer 2 Report

Thank you for the updates to the paper.